# DNN-based Topology Optimisation:
# Spatial Invariance and Neural Tangent Kernel

**Benjamin Dupuis**
Chair of Statistical Field Theory
Ecole Polytechnique Fédérale de Lausanne
Lausanne, Switzerland
`benjamin.dupuis@epfl.ch`

**Arthur Jacot**
Chair of Statistical Field Theory
Ecole Polytechnique Fédérale de Lausanne
Lausanne, Switzerland
`arthur.jacot@epfl.ch`

## Abstract

We study the Solid Isotropic Material Penalisation (SIMP) method with a density field generated by a fully-connected neural network, taking the coordinates as inputs. In the large width limit, we show that the use of DNNs leads to a filtering effect similar to traditional filtering techniques for SIMP, with a filter described by the Neural Tangent Kernel (NTK). This filter is however not invariant under translation, leading to visual artifacts and non-optimal shapes. We propose two embeddings of the input coordinates, which lead to (approximate) spatial invariance of the NTK and of the filter. We empirically confirm our theoretical observations and study how the filter size is affected by the architecture of the network. Our solution can easily be applied to any other coordinates-based generation method.

## 1 Introduction

Topology optimisation [4], also known as structural optimisation, is a method to find optimal shapes subject to some constraints. It has been widely studied in the field of computational mechanics. Here we are interested in the particular case of the Solid Isotropic Material Penalisation (SIMP) method [18, 1], which is a very common method in this field.

Recently some authors have used Deep Neural Networks (DNNs) to perform topology optimisation. We can differentiate two different approaches in the use of DNNs with SIMP. The first approach consists in generating with the classical algorithms a dataset of optimised shapes and train a DNN on this dataset to produce new optimal shapes [3, 31]. Variations of this approach use Generative Adversarial Networks (GAN) [21, 28] to effectively reproduce classical topology optimisation.

In the second approach, the density is generated pointwise by a DNN, which is trained with gradient descent to optimise the density field with respect to the physical constraints, as proposed in [11] to use the power of deep models without giving up exact physics. We focus on the approach of [7, 6] where the density field is generated by a Fully-Connected Neural Network (FCNN) taking the coordinates of a grid as inputs. Surprisingly, [7] observes that the DNN-generated density fields do not feature checkerboard artifacts, which are common in vanilla SIMP. A traditional method to avoid checkerboard patterns is to add a filter [29, 5], but it is not needed for DNN-generated density fields.

In this paper, we analyse theoretically how the use of a DNN to generate the density field affects the learning. Our main theoretical tool is the Neural Tangent kernel (NTK) introduced in [13] to describe the dynamics of wide neural networks [13, 2, 17, 12].

While this paper focuses on linear elasticity and SIMP, our analysis can extended to other physical problems such as heat transfer [19], or any model where an image is generated by a DNN taking the pixel coordinates as inputs (like in [20]).

35th Conference on Neural Information Processing Systems (NeurIPS 2021).

## 1.1 Our contribution

In this paper we study topology optimisation with neural networks. The physical density is represented by a neural network taking an embedding of spatial coordinates as inputs, i.e. the density at a point $x \in \mathbb{R}^d$ is given by $f_\theta(\varphi(x))$ for $\theta$ the parameters of the network and $\varphi$ an embedding. We use theoretical tools, in particular the Neural Tangent Kernel (NTK), to understand how the architecture and hyperparameters of the network affect the optimisation of the density field:

- We show that in the infinite width limit (when the number of neurons in the hidden layers grows to infinity), topology optimisation with a DNN is equivalent to topology optimisation with a density filter equal to the "square root" of the NTK. Filtering is a commonly used technique in topology optimisation, aimed to remove checkerboard patterns.

- In topology optimisation as in other physical optimisation problems, it is crucial to guarantee some spatial invariance properties. If the coordinates are taken as inputs of the network directly, the NTK (and the corresponding filter) is not translation invariant, leading to non-optimal shapes and visual artifacts. We present two methods to ensure the spatial invariance of the NTK: embedding the coordinates on the (hyper-)torus or using a random Fourier features embedding (similar to [32]).

- In traditional topology optimisation, the filter size must be tuned carefully. When optimising with a DNN, the filter size depends on the embedding of the coordinates and the architecture of the network. We define a filter radius for the NTK, which plays a similar role as the classical filter size and discuss how it is affected by the choice of embedding, activation function, depth and other hyperparameters like the importance of bias in the network. This tradeoff can also be analysed in terms of the spectrum of the NTK, explaining why neural networks naturally avoid checkerboard patterns.

We confirm and illustrate these theoretical observations with numerical experiments. Our implementation of the algorithm will be made public at `https://github.com/benjiDupuis/DeepTopo`.

## 2  Presentation of the method

In this paper, we use a DNN to generate the density field used by the Solid Isotropic Material Penalisation (SIMP) method. Our implementation of SIMP is based on [1] and [18]. In this section we introduce the traditional SIMP method and our neural network setting.

### 2.1  SIMP method

We consider a regular grid of $N$ elements where the density of element $i$ is denoted $y_i \in [0, 1]$, informally the value $y_i$ represents the presence of material at a point $i$. Our goal is to optimise over the density $y \in \mathbb{R}^N$ to obtain a shape that can withstand forces applied at certain points, represented by a vector $F$.

The method uses finite element analysis to define a stiffness matrix $K(y) \in S_N^{++}(\mathbb{R})$ from the density $y$ and computes the displacement vector $U(y)$ (which represent the deformation of the shape at all points $i$ as a result of the applied forces $F$) by solving a linear system $K(y)U(y) = F$. In our implementation, we performed it either by using sparse Cholesky factorisation [9, 8] or BICGSTAB method [33] (this last one can be used for a high number of pixels).

The loss function is then defined as the compliance $C(y) = U(y)^T K(y) U(y)$, under a volume constraint of the form $\sum_{i=1}^{N} y_i = V_0$, with $0 \le V_0 \le N$ (see [1, 18]).

### 2.2  A modified SIMP approach

Several methods exist to optimise the density field $y \in \mathbb{R}^N$, such as gradient descent or the so-called Optimality Criteria (OC) [34]. We propose here an optimisation method inspired from [11] which we will refer as the Modified Filtering method (MF). The advantage of this method is that it can be used with or without DNNs, hence allowing comparison between these two approaches. We first present here the model without DNNs.

In our method, the densities $y_i^{\text{MF}}$ are given by:

$$\forall i \in \{1,...,N\}, \; y_i^{\text{MF}} = \sigma(x_i + \bar{b}(X)), \quad \text{with } \bar{b}(X) \text{ such that } \sum_{i=1}^{N} y_i^{\text{MF}} = V_0, \tag{1}$$

for $X = (x_1, ..., x_N) \in \mathbb{R}^N$ and the sigmoid $\sigma(x) = \frac{1}{1+e^{-x}}$. We will denote this operation as: $Y^{\text{MF}} = \Sigma(X)$. The sigmoid ensures that densities are in $[0,1]$ and the choice of the optimal bias $\bar{b}(X)$ ensures that the volume constraint is satisfied.

**Filtering:** If the vector $X$ is optimised directly with gradient descent, SIMP often converges toward checkerboard patterns, i.e. some high frequency noise in the image, which is a common issue with SIMP [1]. To overcome this issue a common technique is to use filtering [29]. In this paper, we consider low-pass density filters of the form: $X = T\bar{X}$ where $T$ represents a convolution on the grid, $\bar{X}$ are the design variables and $X$ is the vector in equation 1. The loss function of this method is then naturally defined as: $\bar{X} \longmapsto C(\Sigma(T\bar{X}))$.

The gradient $\nabla_Y C$ is easily obtained by the self-adjointness of the variational problem [34, 15]. We recover $\nabla_X C$ from $\nabla_Y C$ using an implicit differentiation technique [10]. The following proposition is a consequence of implicit function theorem and chain rules:

**Proposition 2.1.** *Let $\dot{S}$ be the vector with entries $\dot{\sigma}(x_i + \bar{b}(X))$. We have $\nabla_X C = D_X \nabla_Y C$ with:*

$$D_X := -\frac{1}{|\dot{S}|_1} \dot{S}\dot{S}^T + Diag(\dot{S}). \tag{2}$$

*where $|.|_1$ denotes the $l^1$ norm of a vector. Furthermore $D_X$ is a symmetric positive semi-definite matrix whose null-space is the space of constant vectors and has eigenvalues smaller than $\frac{1}{4}$.*

### 2.3 Proposed algorithm: SIMP with Neural networks

Fully-Connected Neural Networks (FCNN) are characterised by the number of layers $L + 1$, the numbers of neurons in each layer $(n_0, n_1, ..., n_L)$ and an activation function $\mu : \mathbb{R} \longrightarrow \mathbb{R}$, here we will use the particular case $n_L = 1$. The activations $a^l \in \mathbb{R}^{n_l}$ and preactivations $\tilde{a}^l \in \mathbb{R}^{n_l}$ are defined recursively for all layers $l$, using the so-called NTK parameterisation [13]:

$$a^0(x) = x, \quad \tilde{a}^{l+1}(x) = \frac{\alpha}{\sqrt{n_l}} W^l a^l(x) + \beta b^l, \quad a^{l+1}(x) = \mu(\tilde{a}^{l+1}(x)), \tag{3}$$

for some hyperparameters $\alpha, \beta \in [0,1]$ representing the contribution of the weights and bias terms respectively. The parameters $\theta = (\theta_p)_p$, consisting in weight matrices $W^l$ and bias vectors $b^l$ are drawn as i.i.d. standard normal random variables $\mathcal{N}(0,1)$. We denote the output of the network as $f_\theta(x) = \tilde{a}^L(x)$.

**Remark:** To ensure that the variance of the neurons at initialization is the equal to 1 at all layers, we choose $\alpha$ and $\beta$ such that $\alpha^2 + \beta^2 = 1$ and use a standardised non-linearity, i.e. $\mathbb{E}_{X \sim \mathcal{N}(0,1)}[\mu(X)^2] = 1$ ([14]).

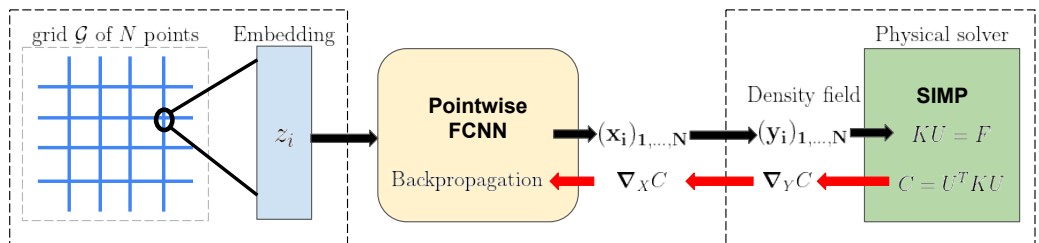

Figure 1: Illustration of our method

In our approach, the pre-densities $X^{\text{NN}}(\theta) = (x_1^{\text{NN}}, ..., x_N^{\text{NN}})$ are generated by a neural network as $x_i^{\text{NN}} = f_\theta(z_i)$ where $z_i \in \mathbb{R}^{n_0}$ is either the coordinates of the grid elements (in this case $n_0 = d$) or

an embedding of those coordinates. We then apply the same transformation $\Sigma$ to obtain the density field $Y^{\text{NN}}(\theta) = \Sigma(X^{\text{NN}}(\theta))$. Our loss function is then defined as:

$$\theta \longmapsto C(Y^{\text{NN}}(\theta)) = C\big(\Sigma(X(\theta))\big).$$

The design variables are now the parameters $\theta$ of the network. The gradient $\nabla_\theta C$ w.r.t. to the parameters is computed by first using Proposition 2.1 to get $\nabla_{Y^{\text{NN}}} C$ followed by traditional backpropagation.

**Remark:** Note the absence of filter $T$ in the above equations, indeed we will show how neural networks naturally avoid checkerboard patterns, making the use of filtering obsolete.

**Initial density field:** The SIMP method is usually initialised with a constant density field [1]. Since the neural network is initialized randomly, the initial density field is random and non-constant. To avoid this problem, we subtract the initial density field and add a well-chosen constant:

$$\forall i \in \{1, ..., N\}, \ x_i(\theta) = \bar{f}_{\theta(t)}(z_i) = f_{\theta(t)}(z_i) - f_{\theta(t=0)}(z_i) + \log\left(\frac{V_0}{N - V_0}\right). \qquad (4)$$

We used equation 4 to compute $X(\theta)$ in our numerical experiments.

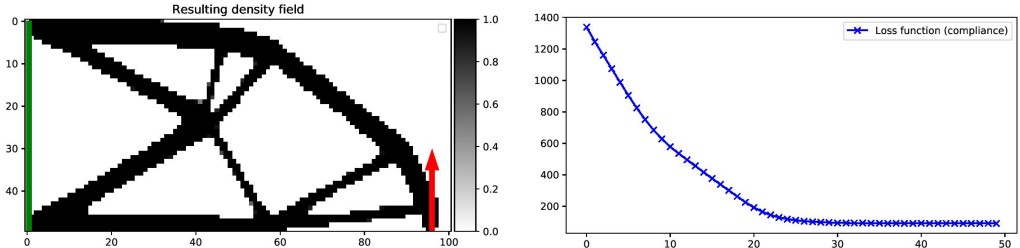

Figure 2: Example of result of our method with applied forces (red arrow) and a fixed boundary (green). Here we used a Gaussian embedding (see section 4 for details).

## 3 Theoretical Analysis

### 3.1 Analogy between the Neural Tangent Kernel and filtering techniques

In our paper, we use the Neural Tangent Kernel (NTK [13]) as the main tool to analyse the training behaviour of the FCNN. In our setting (where $n_L = 1$) the NTK is defined as:

$$\forall z, z' \in \mathbb{R}^{n_0}, \ \Theta_\theta^L(z, z') = \sum_p \frac{\partial f_\theta}{\partial \theta_p}(z)\frac{\partial f_\theta}{\partial \theta_p}(z') = (\nabla_\theta f_\theta(z) | \nabla_\theta f_\theta(z')).$$

This is a positive semi-definite kernel. Given some inputs $z_1, ..., z_N$ we define the NTK Gram matrix as: $\tilde{\Theta}_\theta^L := \big(\Theta^L(z_i, z_j)\big)_{1 \le i,j \le N} \in \mathbb{R}^{N \times N}$.

Assuming a small enough learning rate, the evolution of the network under gradient descent is well approximated by the gradient flow dynamics $\partial_t \theta(t) = -\nabla_\theta C(t)$. The evolution of the output of the network $X^{\text{NN}}(\theta)$ can then easily be expressed in terms of the NTK Gram matrix [14] for a loss $\mathcal{L}$:

$$\partial_t X^{\text{NN}}(\theta(t)) = -\tilde{\Theta}_{\theta(t)}^L \nabla_{X^{\text{NN}}} \mathcal{L}.$$

From this equation we can derive the evolution of the physical density field $Y^{\text{NN}}$ in our algorithm:

**Proposition 3.1.** *If the network is trained under this gradient flow, then by applying chain rules, we can prove that the density field follows the equation:*

$$\partial_t Y^{NN}(\theta(t)) = -D_X(t)\tilde{\Theta}_{\theta(t)}^L D_X(t)\nabla_Y C(Y^{NN}(\theta(t))). \qquad (5)$$

The analogy between the NTK and filtering techniques comes from the following observation. With Modified Filtering with a filter $T$, we show similarly that the density field $Y^{\text{MF}}$ evolves as

$$\partial_t Y^{\text{MF}}(t) = -D_X(t)TT^T D_X(t)\nabla_Y C(Y^{\text{MF}}(t)). \qquad (6)$$

We see that the NTK Gram matrix and the squared filter $TT^T$ play exactly the same role. An important difference however is that the NTK is random at initialisation and evolves during training.

This difference disappears for large widths (when $n_1, \ldots, n_{L-1}$ are large), since the NTK converges to a deterministic and time independent limit $\tilde{\Theta}_\infty^L$ as $n_1, \ldots, n_{L-1} \to \infty$ [13]. Furthermore, in contrast to the finite width NTK (also called empirical NTK), we have access to a closed form formula for the limiting NTK $\tilde{\Theta}_\infty^L$ (given in the appendix).

In the infinite width limit, the evolution of the physical densities is then expressed in terms of the limiting NTK Gram matrix $\tilde{\Theta}_\infty^L$:

$$\partial_t Y^{\mathrm{NN}}(\theta(t)) = -D_X(t)\tilde{\Theta}_\infty^L D_X(t)\nabla_Y C(Y^{\mathrm{NN}}(\theta(t))). \tag{7}$$

From now on we will focus on this infinite-width limit, comparing the NTK Gram matrix $\tilde{\Theta}_\infty^L$ and the squared filter $TT^T$. Recent results [17, 2, 12] suggest that this limit is a good approximation when the width of the network is sufficiently large. For more details see the appendix, where we compare the empirical NTK with its limiting one and plot its evolution in our setting.

## 3.2 Spatial invariance

In physical problems such as topology optimisation, it is important to ensure that certain physical properties are respected by the model. We focus in this section on the translation and rotation invariance of topology optimisation: if the force constraints are rotated or translated, the resulting shape should remain the same (up to rotation and translation), as in Figure 4 (b.1 and b.2).

In Modified Filtering method, this property is guaranteed if the filter $T$ is translation and rotation invariant. In contrast the limiting NTK is in general invariant under rotation [13] but not translation. As Figure 4 shows, this leads to some problematic artifacts. The NTK can be made translation and rotation invariant by first applying an embedding $\varphi : \mathbb{R}^d \longrightarrow \mathbb{R}^{n_0}$ with the properties that for any two coordinates $p, p'$, $\varphi(p)^T \varphi(p')$ only depends on the distance $\|p - p'\|_2$. Since the rotation invariance of the NTK implies that $\Theta_\infty^L(z, z')$ depends only on the scalar products $z^T z'$, $zz^T$ and $z'z'^T$, we have that $\Theta_\infty^L(\varphi(p), \varphi(p'))$ depends only on $\|p - p'\|$ as needed.

The issue is that for finite $n_0$ there is no non-trivial embedding $\varphi$ with this property:

**Proposition 3.2.** *Let* $\varphi : \mathbb{R}^d \to \mathbb{R}^{n_0}$ *for* $d > 2$ *and any finite* $n_0$. *If* $\varphi$ *satisfies* $\varphi(x)^T \varphi(x') = K(\|x - x'\|)$ *for some continuous function* $K$ *then both* $\varphi$ *and* $K$ *are constant.*

To overcome this issue, we present two approaches to approximate spatial invariance with finite embeddings: an embedding on a (hyper)-torus and a random feature [23] embedding based on Bochner theorem [26].

### 3.2.1 Embedding on a hypertorus

In this subsection we consider the following embedding of a $n_x \times n_y$ regular grid on a torus:

$$\mathbb{R}^2 \ni p = (p_1, p_2) \longmapsto \varphi(p) = r(\cos(\delta p_1), \sin(\delta p_1), \cos(\delta p_2), \sin(\delta p_2)), \tag{8}$$

where $\delta > 0$ is a discretisation angle (our default choice is $\delta = \frac{\pi}{2\max(n_x, n_y)}$). One can use similar formulas for $d > 2$ (leading to an hyper-torus embedding), we used $d = 2$ in equation 8 for simplicity.

This embedding leads to an exact translation invariance and an approximate rotation invariance:

$$\varphi(p)^T \varphi(p') = r^2(\cos(\delta(p_1 - p_1')) + \cos(\delta(p_2 - p_2'))) = r^2\left(2 - \frac{\delta^2}{2}\|p - p'\|_2^2\right) + \mathcal{O}(\delta^4\|p - p'\|_4^4).$$

As a result, the limiting NTK $\Theta_\infty(\varphi(p), \varphi(p'))$ is translation invariant and approximately rotation invariant (for small $\delta$ and/or when $p, p'$ are close to each other). Moreover, if we look at the limiting NTK on the whole torus, we obtain that the gram matrix $\tilde{\Theta}_\infty$ is a discrete convolution on the input grid, with nice properties summed up in the following proposition:

**Proposition 3.3.** *We can always extend our* $n_x \times n_y$ *grid and choose* $\delta$ *such that the embedded grid covers the whole torus (typically* $\delta = \frac{\pi}{2\max(n_x, n_y)}$ *and take a* $n \times n$ *grid with* $n = 4\max(n_x, n_y)$).

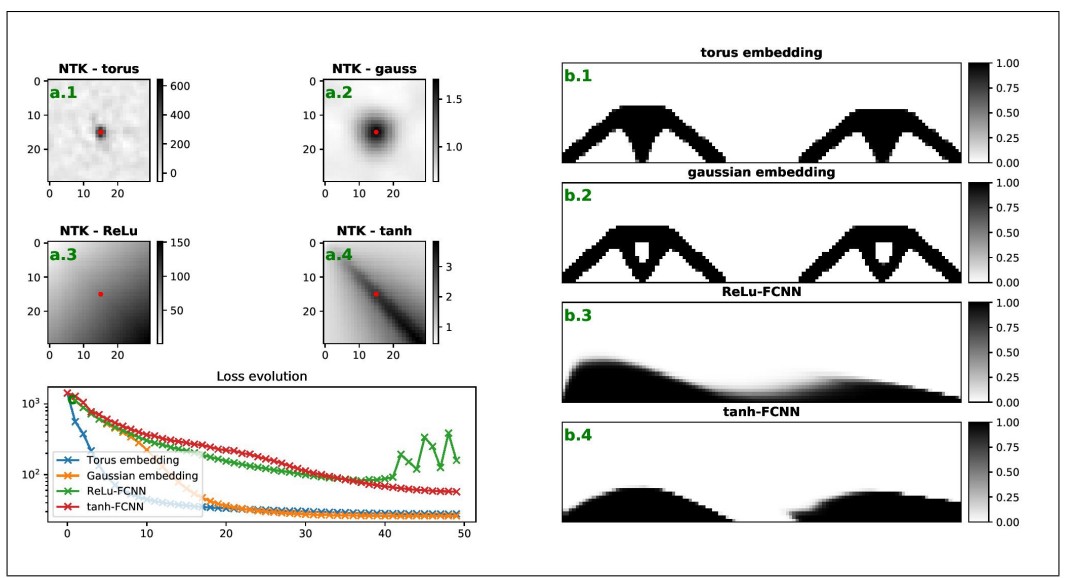

Figure 4: Left: empirical NTK of FCNNs with both embedding (a.1, a.2, see Section 4.1 for details) or without embedding (a.3 with ReLu, a.4 with tanh). Right: Corresponding shape obtained after training. Note that methods without spatial invariance particularly struggles with this symmetric load case (b.3, b.4) while both "embedded methods" respect the symmetry (b.1, b.2). We also observed that training with non-embedded methods is very unstable

*Then the Gram matrix $\tilde{\Theta}_\infty$ of the limiting NTK is a 2D discrete convolution matrix. Moreover the NTK Gram matrix has a positive definite square root $\sqrt{\tilde{\Theta}_\infty}$ which is also a discrete convolution matrix.*

As we know, the eigenvectors of such a convolution matrix are the 2D Fourier vectors. The corresponding eigenvalues are the discrete Fourier transforms of the convolution kernel.

The square root of the NTK Gram matrix $\sqrt{\tilde{\Theta}_\infty}$ then corresponds to the filtering matrix $T$ in our analogy. Figure 3 shows that on the full torus, the matrix square root $\sqrt{\tilde{\Theta}_\theta}$ indeed looks like a typical smoothing filter.

As Figure 4 shows, the torus embedding method gives good numerical results and respect the symmetry of the applied forces $F$.

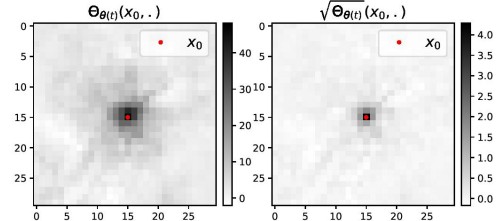

Figure 3: Representation of one line of $\tilde{\Theta}_\theta$ on the full torus and of its square root. We used $\beta = 0.2$ and $\omega = 3$ (see Section 4.1) here to make the filter visible on the whole torus.

### 3.2.2 Random embeddings for radial kernels

Another approach to approximate a rotation and translation invariant embedding is to use random Fourier features [23], which is a general method to approximate shift invariant kernels of the form $k(x, y) = k(x - y)$. By Bochner theorem [26], any continuous non-zero radial kernel $k(x - y) = K(\|x - y\|)$ can be written as the the (scaled) Fourier transform of a probability measure $\mathbb{Q}$ on $\mathbb{R}^d$:

$$k(r) = k(0) \int_{\mathbb{R}^d} e^{i\omega \cdot r} d\mathbb{Q}(\omega).$$

For radial kernels, we formulate random Fourier features embeddings $\varphi : \mathbb{R}^d \to \mathbb{R}^{n_0}$ as follows:

$$\varphi(p)_i = \sqrt{2k(0)} \sin(w_i^T p + \frac{\pi}{4} + b_i),$$

for i.i.d. samples $w_1, ..., w_{n_0} \in \mathbb{R}^d$ from $\mathbb{Q}$ (which is also invariant by rotation) and i.i.d. samples $b_1, ..., b_{n_0} \in \mathbb{R}^d$ from any symmetric probability distribution (or uniform laws on $[0, 2\pi]$). By the law of large numbers for large $n_0$, we have the approximation $\frac{1}{n_0}\varphi(p)^T\varphi(p') \simeq k(p - p')$.

**Gaussian embedding:** Depending on the kernel $k$ that we want to approximate, it may be difficult to sample from the distribution $\mathbb{Q}$. The simplest case is for a Gaussian kernel $k(d) = e^{-\frac{1}{2\ell^2}d^2}$, where the distribution $\mathbb{Q}$ of the weights $w_i$ is $\mathcal{N}(0, \frac{1}{\ell^2}I_d)$, i.e. the entries $w_{ij}$ are all i.i.d. $\mathcal{N}(0, \frac{1}{\ell^2})$ Gaussians. For this reason this is the embedding that we will use in our numerical experiments. Note the similarity between this type of embedding and an untrained first layer of a FCNN with sine activation function, weights $w_i$ and bias $b_i$.

Moreover, the following result shows that we can still define a "square root" of the NTK with those types of embedding and thus complete the analogy with equation 6.

**Proposition 3.4.** *Let $\varphi$ be an embedding as described above for a positive radial kernel $k \in L^1(\mathbb{R}^d)$ with $k(0) = 1$, $k \geq 0$. Then there is a filter function $g : \mathbb{R} \to \mathbb{R}$ and a constant $C$ such that for all $p, p'$:*

$$\lim_{n_0 \to \infty} \Theta_\infty(\varphi(p), \varphi(p')) = C + (g \star g)(p - p'), \tag{9}$$

*where $\Theta_\infty$ is the limiting NTK of a network with a Lipschitz, non-constant and standardised activation function $\mu$. (Here $\star$ denotes the convolution product).*

As the matrix $D_X$ in equation 7 cancels out the constant frequency (proposition 2.1), the constant $C$ doesn't matter, i.e. $D_X \tilde{\Theta}_\infty^{(L)} D_X = D_X \left( \tilde{\Theta}_\infty^{(L)} - C \right) D_X$.

## 4 Experimental analysis

### 4.1 Setup

Most of our experiments were conducted with a torus embedding or a gaussian embedding. For the SIMP algorithm, we adapted the code described in [1, 18]. Here are the hyperparameters used in the experiments.

For the Gaussian embedding, we used $n_0 = 1000$ and a length scale $\ell = 4$. This embedding was followed by one hidden linear layer of size 1000 with standardized ReLu ($x \mapsto \sqrt{2}\max(0, x)$) and a bias parameter $\beta = 0.5$.

For the torus embedding we set the torus radius to $r = \sqrt{2}$ (to be on a standard sphere) and the discretisation angle to $\delta = \frac{\pi}{2\max(n_x, n_y)}$ (to cover roughly half the torus, which is a good trade-off between rotation invariance and kernel size), where $n_x \times n_y$ is the size of the grid. It was followed by 2 linear layers of size 1000 with $\beta = 0.1$. The ReLu activation is not well-suited in this case because it induces filters that are too wide. The large radius of the NTK kernel can be understood in relation with the order/chaos regimes [27, 22], as observed in [14] the ReLU lies in the ordered regime when $\beta > 0$, leading to a "wide" kernel, a narrower kernel can be achieved with non-linearities which lie in the chaotic regime instead. We used a cosine activation of the form $x \mapsto \cos(\omega x)$, which has the advantage that the width of the filter can be adjusted using the $\omega$ hyperparameter, see Section 7. When not stated otherwise we used $\omega = 5$.

Even though our theoretical analysis is for gradient flow, we obtain similar results with other optimizers such as RPROP [24] (learning rate $10^{-3}$) and ADAM [16] (learning rate $10^{-3}$). RPROP gave the fastest results, possibly because it is well-suited for batch learning [25]. Vanilla gradient descent can be very slow due to the vanishing of the gradients when the image becomes almost binary (due to the sigmoid), we therefore gradually increased the learning rate during training to compensate.

### 4.2 Spectral analysis

In SIMP convolution with a low pass filter ensures that low frequencies are optimised faster than high frequencies, to avoid checkerboards.

With the embeddings proposed in the last two subsections, the limiting NTK takes the form of a convolution over the input space $\mathbb{R}^d$. Figure 5 represents the eigenvalues and eigenimages of the

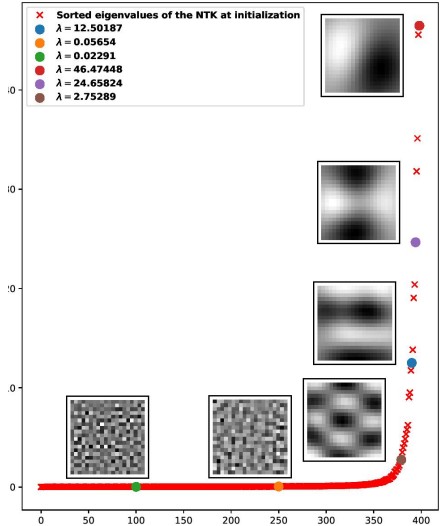

Figure 5: Sorted eigenvalues of the empirical NTK with some eigenvectors (reshaped as images). Obtained with a Gaussian embedding.

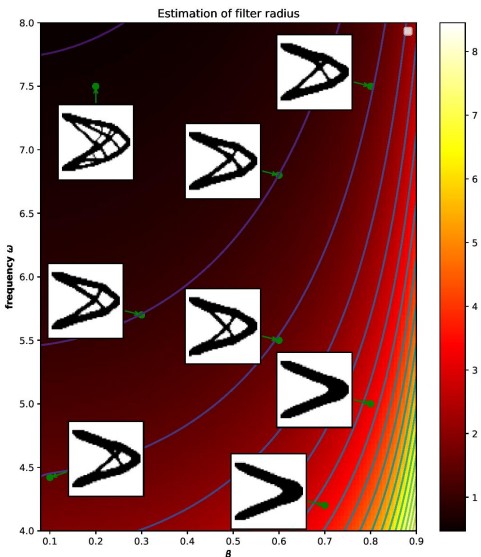

Figure 6: Colormap of $\widehat{R}_{1/2}$ in the $(\beta, \omega)$ plane, torus embedding. Level lines and shapes obtained for different radius are represented.

NTK Gram matrix $\tilde{\Theta}_{\theta(t)}$. Even though this plot is done for a finite width network and a finite random embedding, we see that the eigenimages look like 2D Fourier modes. The fact that the low frequencies have the largest eigenvalues supports the similarity between the NTK and a low pass filter.

This may explain why neural networks naturally avoid checkerboard patterns: the low frequencies of the shape are trained faster than the high frequencies which lead to checkerboard patterns.

### 4.3  Filter radius

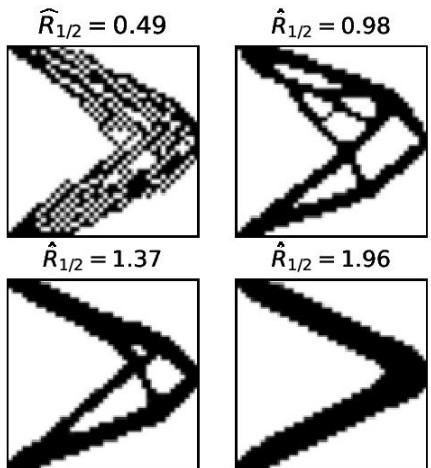

Figure 7: Shape obtained for different values of $\widehat{R}_{1/2}$ with a Gaussian embedding for different values of $\ell \in \{0.5, 1, 1.4, 2\}$.

In the classical SIMP algorithm, the choice of the radius of the filter $T$ is critical. It controls the appearance of checkerboards or intermediate densities.

When using DNNs, there is no explicit choice of filter radius, since the filter depends on the embedding and the architecture of the network. In Section 3.2 we have shown that the NTK is approximately invariant, it can hence be expressed as:

$$\Theta^L_{\theta(t)}(\varphi(p), \varphi(p')) \simeq \Phi_\infty(\|p - p'\|),$$

where $\Phi_\infty$ can be analytically expressed with the embedding and the limiting NTK (see appendix for a detailed example).

The kernels we consider do not have compact support in general, we therefore focus instead on the radius at half-maximum of $\Phi_\infty$:

$$\Phi_\infty(\widehat{R}_{1/2}) = \frac{1}{2}\big(\Phi_\infty(0) + \inf_r \Phi_\infty(r)\big).$$

Note that for simplicity we are computing here the radius of the squared filter, since obtaining a closed form formula for the square root of the NTK is more difficult. For Gaussian filters the radius of the squared filter is $\sqrt{2}$ times that of the original, suggesting that the filter radius is well estimated by $\frac{1}{\sqrt{2}}\widehat{R}_{1/2}$.

The quantity $\widehat{R}_{1/2}$ is a function of the hyperparameters of the network ($\alpha, \beta, L$, see appendix) and of the embedding (the lengthscale $\ell$). Using the formula for $\widehat{R}_{1/2}$, these hyperparameters can be tuned to obtain a specific filter radius.

With the Gaussian embedding, the radius of the filter can easily be adjusted by changing the lengthscale $\ell$ of the embedding. As illustrated in Figure 7.

With the torus embedding, we instead have to change the hyperparameters of the network to adjust the radius of the filter. With the ReLU activation function, the radius is very large which makes it impossible to obtain precise shape. The solution we found is to use a cosine activation $x \mapsto \cos(\omega x)$ with hyper-parameter $\omega$. Figure 6 shows how the radius decreases as $\omega$ increases. The $\beta$ parameter has the opposite effect, as increasing it increases the radius. For different values of $\omega$ and $\beta$, we obtain a variety of radius and plot the resulting shapes. This plot also illustrates the role of the radius in the determination of the resulting shape. The fact that cosine activation leads to an adjustable NTK radius could explain why periodic activation function help in the representation of high frequency signal as observed in [30].

The effect of depth is more complex. For large depths $L$ the NTK either approaches a constant kernel in the so-called order regime (with infinite radius) or a Kronecker delta kernel in the so-called chaos regime (with zero radius) [22, 27, 14]. Depending on whether we are in the order or chaos regime (which is determined by the activation function $\mu$ and the parameters $\alpha, \beta$), increasing the depth can either increase or decrease the radius.

We conducted an experimental study of the influence of this parameter on the geometry of the final shape. We observed that its complexity (number of holes, high frequencies) is highly controlled by $\widehat{R}_{1/2}$. We see in Figure 7 and 6 some examples of shape obtained for several values of $\widehat{R}_{1/2}$.

## 4.4 Up-sampling

Since the density field is generated by a DNN, it can be evaluated at any point in $\mathbb{R}^d$, hence allowing upsampling. As Figure 8 shows, with our method we obtain a smooth and binary shape. Something interesting happens when the network is trained without an embedding: when upsampling we observe some visual artifacts plotted in Figure 9. We believe that it is due to the lack of spatial invariance.

Note that this second experiment was done with batch norm, as described in [7], since for this problem it was difficult to obtain a good shape with a vanilla ReLU-FCNN. With our embeddings, we can achieve complex shapes without batch-norm.

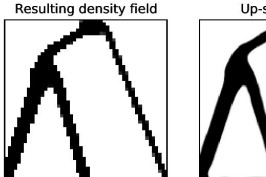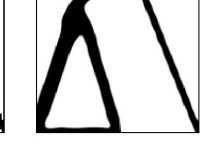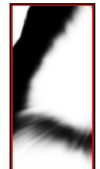

Figure 8: Density field obtained with a Torus embedding (left) and up sampling of factor 6 of the same network (right).

Figure 9: Exemple of up-sampling of a FCNN (ReLu FCNN with batchnorms) without embedding, exhibing typical visual artifacts.

## 5 Conclusion

Using the NTK, we were able to give a simple theoretical description of topology optimisation with DNNs, showing a similarity to traditional filtering techniques. This theory allowed us to identify a problem: since the NTK is not translation invariant, the spatial invariance of topology optimisation is not respected, leading to visual artifacts and non-optimal shapes. We propose a simple solution to this problem: adding a spatial invariant embedding to the coordinates before the DNN.

Using this method, our models are able to learn efficient shapes while avoiding checkerboard patterns. We give tools to adjust the implicit filter size induced by the hyperparameters, to give control over the complexity of the final shape. Using the learned network, we can easily perform good quality

up-sampling. The techniques described in this paper can easily be translated to any other problem where spatial invariance is needed.

The NTK is a simple yet powerful tool to analyse a practical method such as SIMP when combined with a DNN. Morover it can be used to make informed choices of the DNN's architecture and hyperparameters.

## Acknowledgments and Disclosure of Funding

There is no funding or competing interests associated to this work.

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
