# DNN-based Topology Optimisation:
# Spatial Invariance and Neural Tangent Kernel
# Supplementary Material

**Benjamin Dupuis**
Chair of Statistical Field Theory
Ecole Polytechnique Fédérale de Lausanne
Lausanne, Switzerland
benjamin.dupuis@epfl.ch

**Arthur Jacot**
Chair of Statistical Field Theory
Ecole Polytechnique Fédérale de Lausanne
Lausanne, Switzerland
arthur.jacot@epfl.ch

## A  Derivation of the algorithm

In this section we show how to derive the equations used in our algorithm, especially the ones corresponding to implicit differentiation [2]. Let us recall that we consider a vector $X \in \mathbb{R}^N$ and compute a vector $Y = \Sigma(X) \in [0,1]^N$ (either $Y^{\text{MF}}$ or $Y^{\text{NN}}$) by:

$$\forall i \in \{1, ..., N\},\ y_i = \sigma(x_i + \bar{b}(X)), \quad \text{such that: } \sum_{i=1}^{N} y_i = V_0, \quad \sigma(x) = \frac{1}{1 + e^{-x}},$$

Where $X$ denotes $(x_1, ..., x_N)$.

We want to show that this operation is well defined and find a formula to recover $\nabla_X C$ from a given $\nabla_Y C$. More precisely we have the following result.

**Proposition A.1** (Proposition 2.1 in the paper). *Let $X \in \mathbb{R}^N$, the operation $Y = \Sigma(X)$ is well defined. Moreover, let $\dot{S}$ be the vector of the $\dot{\sigma}(x_i + \bar{b}(X))$. Then we have $\nabla_X C = D_X \nabla_Y C$ with:*

$$D_X := -\frac{1}{|\dot{S}|_1} \dot{S}\dot{S}^T + Diag(\dot{S}). \tag{1}$$

*$D_X$ is a symmetric positive semi-definite matrix whose kernel corresponds to constant vectors and has eigenvalues smaller than $\frac{1}{2}$.*

*Proof:* Let us consider the function $F : \mathbb{R}^N \times \mathbb{R} \longrightarrow \mathbb{R}$ defined by: $F(z, b) = \sum_{i=1}^{N} \sigma(z_i + b)$. It is clear that $F(X, .)$ is stricty increasing on $\mathbb{R}$ from 0 to $N$. Then $\exists! \bar{b} \in \mathbb{R}$ such that $F(X, \bar{b}) = V_0$.

As $\partial_b F(X, \bar{b}) > 0$, by the implicit functions theorem, there exists a neighbourhood $V$ of $X$ in $\mathbb{R}^N$, a neighbourhood $U$ of $\bar{b}$ in $\mathbb{R}$ and a function $\bar{b} : V \longrightarrow \mathbb{R}$ of class $\mathcal{C}^1$ such that:

$$\forall (z, b) \in V \times U,\ F(z, b) = V_0 \iff b = \bar{b}(z).$$

Moreover we also get from the implicit function theorem that:

$$\frac{\partial \bar{b}}{\partial x_i}(X) = -\left(\frac{\partial F}{\partial b}(X, \bar{b})\right)^{-1} \frac{\partial F}{\partial x_i}(X, \bar{b}) = -\left(\sum_{j=1}^{N} \dot{\sigma}(x_j + \bar{b})\right)^{-1} \dot{\sigma}(x_i + \bar{b}),$$

and we can apply chain rules:

$$\frac{\partial C}{\partial x_i} = \sum_{j=1}^{N} \frac{\partial C}{\partial y_j} \frac{\partial y_j}{\partial x_i}$$

$$= \sum_{j=1}^{N} \frac{\partial C}{\partial y_j} \dot{\sigma}(x_j + \bar{b}(x))\left(\frac{\partial \bar{b}}{\partial x_i} + \delta_{ij}\right),$$

35th Conference on Neural Information Processing Systems (NeurIPS 2021).

So that equation 1 makes sense. Now, if we denote $\dot{S} = (a_1, ..., a_N)$, let us recall that we defined $a_i = \dot{\sigma}x_i + \bar{b}(X)$ where $\sigma$ is the sigmoid function. By taking any $u \in \mathbb{R}^N$, we remark that:

$$\left(D_X u\right)_i = \frac{a_i}{|\dot{S}|_1} \sum_{j=1}^{N} a_j(u_i - u_j). \tag{2}$$

We easily deduce from equation 2 that $\ker(D_X) = \mathrm{span}(1_N)$ and that $D_X \in S_N^+(\mathbb{R})$. Indeed:

$$\forall u \in \mathbb{R}^N, \quad u^T(D_X)u = -\frac{1}{|\dot{S}|_1}u^T \dot{S}\dot{S}^T u^T + \sum_{i=1}^{N} a_i u_i^2$$

$$= \frac{1}{|\dot{S}|_1}\left\{ -\left(\sum_{i=1}^{N} a_i u_i\right)^2 + \left(\sum_{i=1}^{N} a_i\right)\left(\sum_{i=1}^{N} a_i u_i^2\right)\right\}$$

$$= \frac{1}{|\dot{S}|_1} \sum_{1 \le i,j \le N} a_i a_j u_i(u_i - u_j)$$

$$= \frac{1}{|\dot{S}|_1} \sum_{1 \le i < j \le N} a_i a_j(u_i - u_j)^2 \ge 0.$$

**Eigenvalues**: We already know that 0 is an eigenvalue with multiplicity 1. So let $u \ne 0$ in $\mathbb{R}^N$ and $\lambda > 0$ such that: $D_X u = \lambda u$. Then we easily show:

$$\forall i \in [\![1, N]\!], \quad \frac{a_i - \lambda}{a_i}u_i = \frac{1}{|\dot{S}|_1}\sum_{j=1}^{N} a_j u_j =: \langle u\rangle_a.$$

If $\langle u\rangle_a = 0$, then necessarily $\lambda \in \{a_1, ..., a_N\}$
If $\langle u\rangle_a \ne 0$, then we can assume (by normalising $u$) that $\langle u\rangle_a = 1$ and we have $u_i = \frac{a_i}{a_i - \lambda}$. Then we can replace $u_i = \frac{a_i}{a_i - \lambda}$ in the equation $\langle u\rangle_a = 1$:

$$\sum_{j=1}^{N} a_j = \sum_{j=1}^{N} \frac{a_j^2}{a_j - \lambda}, \quad \text{which by substraction leads to} \quad F(\lambda) := \sum_{j=1}^{N} \frac{a_j}{a_j - \lambda} = 0,$$

By studying the function $F$, we see that $\forall \lambda > \max_i(a_i), \ F(\lambda) < 0$. Therefore an eigenvalue always satisfies the inequality:

$$\lambda \le \max\{a_1, ..., a_N\} \le \|\dot{\sigma}\|_\infty = \frac{1}{4},$$

The last inequality coming from the fact that $a_i = \dot{\sigma}(x_i + \bar{b}(X))$, as mentionned earlier.

**Remark:** As shown above an important property of the matrix $D_X$ is that it cancels out constants, which allows us to consider the limiting NTK up to some constant. The fact that the eigenvalues of $D_X$ are in $[0, \frac{1}{4}]$ can help to avoid exploding gradients.

## B   Equations of evolution

We quickly show how equations 5, 6 and 7 of the paper are derived. The proofs are mainly based on chain rules.

Let us first remark that the matrix $D_X$ introduced above actually corresponds to the jacobian matrix $\nabla_X \Sigma$ of the application $\Sigma : \mathbb{R}^N \longrightarrow [0, 1]^N$. So we can immediately applied chain rules to $Y^{\mathrm{NN}} = \Sigma(X(\theta))$ and get:

$$\frac{\partial Y^{\mathrm{NN}}}{\partial t} = D_{X(\theta(t))}\frac{\partial X(\theta(t))}{\partial t}$$

$$= -D_{X(\theta(t))}\tilde{\Theta}_{\theta(t)}^{L}\nabla_{X_{\theta(t)}}C \quad \text{(Gradient Descent)}$$

$$= -D_{X(\theta(t))}\tilde{\Theta}_{\theta(t)}^{L}D_{X(\theta(t))}\nabla_{Y^{\mathrm{NN}}}C(\theta(t)) \quad \text{(By proposition A.1).}$$

Similarly, for the MF method, we set $X = T\bar{X}$ and obtain:

$$\frac{\partial Y^{\mathrm{MF}}}{\partial t} = D_{X(t)} \frac{\partial X(t)}{\partial t}$$

$$= D_{X(t)} T \frac{\partial \bar{X}(t)}{\partial t} \quad \text{(Linearity)}$$

$$= -D_{X(t)} T \nabla_{\bar{X}} C \quad \text{(Gradient descent)}$$

$$= -D_{X(t)} T T^T \nabla_X C \quad \text{(Chain rule)}$$

$$= -D_{X(t)} T T^T D_{X(t)} \nabla_{Y^{\mathrm{MF}}} C.$$

## C   Details about embeddings

### C.1   Torus embedding

The aim of this section is to give details about properties of the limiting NTK in case of Torus embedding. As a reminder we consider the following embedding:

$$\mathbb{R}^2 \ni p = (p_1, p_2) \longmapsto \varphi(p) = r(\cos(\delta p_1), \sin(\delta p_1), \cos(\delta p_2), \sin(\delta p_2));$$

In particular we show the following proposition which basically says that $\tilde{\Theta}_\infty$ is in that case a discrete convolution and derive from there its spectral properties and construct its positive semi-definite square root

**Proposition C.1** (Proposition 3.3 in the paper). *We can always extend our $n_x \times n_y$ grid and choose $\delta$ such that the embedded grid covers the whole torus (typically $\delta = \frac{\pi}{2\max(n_x,n_y)}$ and take a $n \times n$ grid with $n = 4\max(n_x,n_y)$). Then the Gram matrix $\tilde{\Theta}_\infty$ of the limiting NTK is a 2D discrete convolution matrix. Moreover the NTK Gram matrix has a positive definite square root $\sqrt{\tilde{\Theta}_\infty}$ which is also a discrete convolution matrix.*

*proof:* We assume that we extend the grid in a $n \times n$ grid with $n \geq n_x, n_y$. Now we take $\delta = \frac{2\pi}{n}$ and we consider the limiting NTK Gram matrix on $\varphi(\llbracket n, n \rrbracket \times \llbracket n, n \rrbracket)$.

As $\Theta_\infty(\varphi(p), \varphi(p'))$ depends only on $p - p'$, we can see the limiting NTK Gram Matrix as a discrete convolution kernel $\mathcal{K}$ acting on $\mathbb{Z}/n\mathbb{Z} \times \mathbb{Z}/n\mathbb{Z}$:

$$\Theta_\infty((k, k'), (j, j')) = \mathcal{K}(k - k', j - j'),$$

For $(k, k'),\ (j, j') \in \mathbb{Z}/n\mathbb{Z} \times \mathbb{Z}/n\mathbb{Z}$.

We see $\tilde{\Theta}_\infty$ as a $n^2$ square matrix with each index in $\mathbb{Z}/n\mathbb{Z} \times \mathbb{Z}/n\mathbb{Z}$.

We introduce the Fourier vectors $\Omega_m = (e^{-i2\pi \frac{mk}{n}})_{0 \leq k \leq n_x - 1}$. As $\tilde{\Theta}_\infty$ is a 2D convolution matrix, we classically have the following results:

The eigenvectors of $\tilde{\Theta}_\infty$ are exactly given by:

$$\Omega_m \otimes \Omega_M,$$

for $0 \leq m \leq n_x - 1$ and $0 \leq M \leq n_y - 1$, $\otimes$ denotes the Kronecker product. The corresponding eigenvalue is given by the discrete Fourier transform $\widehat{\mathcal{K}}(m, M)$ with:

$$\widehat{\mathcal{K}}(m, M) = \sum_{j=0}^{n-1} \sum_{j'=0}^{n-1} e^{-i2\pi \frac{mj}{n}} e^{-i2\pi \frac{Mj'}{n}} \mathcal{K}(j, j').$$

Moreover, as the matrix $\tilde{\Theta}_\infty$ is positive definite (from the positive definiteness of the NTK, [3]) those eigenvalues verify $\widehat{\mathcal{K}}(m, M) \geq 0$ and it makes sense to write the square root of the NTK Gram Matrix as the inverse Fourier transform of the $\sqrt{\widehat{\mathcal{K}}(m, M)}$:

$$\sqrt{\tilde{\Theta}_\infty}((k, k'), (j, j')) = \frac{1}{n^2} \sum_{m=0}^{n-1} \sum_{M=0}^{n-1} e^{i2\pi \frac{m(j-k)}{n}} e^{i2\pi \frac{M(j'-k')}{n}} \sqrt{\widehat{\mathcal{K}}(m, M)}, \tag{3}$$

It is easy to see that the matrix defined by equation 3 is symmetric and positive semi-definite. Indeed we can write $\sqrt{\tilde{\Theta}_\infty}((k, k'), (j, j')) = g(k - j, k' - j')$ with $g$ the Fourier transform of a positive vector.

Moreover it follows from the (discrete) convolution theorem that $\sqrt{\tilde{\Theta}_\infty}^2 = \tilde{\Theta}_\infty$. Therefore $\sqrt{\tilde{\Theta}_\infty}((k, k'), (j, j'))$ is indeed the positive semi-definite matrix square root of $\tilde{\Theta}_\infty$.

Thus the square root of the NTK Gram matrix can be seen as a convolution filter as well (it is invariant by translation as a function of $(k - j, k' - j')$).

## C.2 Dimension of radial embeddings

In this section we prove that feature maps associated to continuous radial kernels are either trivial or of infinite dimension. this result is what motivates discussion in section 3.2 of the paper.

Let us first recall Bochner theorem ([6]):

**Theorem C.1** (Bochner). *Let $(x, y) \mapsto k(x - y)$ be a continuous shift invariant positive definite kernel on $\mathbb{R}^d$. Then it is the Fourier transform of a finite positive measure $\Lambda$ on $\mathbb{R}^d$:*

$$k(r) = \int_{\mathbb{R}^d} e^{i\omega \cdot r} d\Lambda(\omega).$$

The function $k$ appearing in the above theorem will be called a positive definite function, according to the following definition:

**Definition C.1.** *Let $k : \mathbb{R}^d \longrightarrow \mathbb{R}$, then $k$ is a positive definite function when for all $n$, all $p_1, \ldots, p_n \in \mathbb{R}^d$ and all $c_1, \ldots, c_n \in \mathbb{R}$ we have:*

$$\sum_{1 \le i, j \le n} c_i c_j k(x_i - x_j) \ge 0.$$

Moreover we will denote $SO(d)$ the set of rotations matrices of dimension $d$ and the Fourier transform (for an integrable function $\psi$):

$$\mathcal{F}\psi(\omega) = \int_{\mathbb{R}^p} \psi(p) e^{-i\omega \cdot p} dp.$$

Let us now recall the result that we want to prove:

**Proposition C.2** (Proposition 3.2 in the paper). *Let $\varphi : \mathbb{R}^d \to \mathbb{R}^m$ for $d > 2$ and any finite $m$. If $\varphi$ satisfies*

$$\varphi(x)^T \varphi(x') = K(\|x - x'\|) \tag{4}$$

*for some continuous function $K$ then both $\varphi$ and $K$ are constant. We will denote $k(x - x') := K(\|x - x'\|)$.*

*Proof:* We procede in the following way: We consider an embedding $\varphi$ as described above and we are going to show that, when $K$ is not constant, one can construct arbitrarily big linearly independent families $\varphi(p_1), \ldots, \varphi(p_n)$.

For now let us take pairwise distinct $p_1, \ldots, p_n \in \mathbb{R}^d$ and $c_1, \ldots, c_n \in \mathbb{R}$ such that:

$$\sum_{k=1}^{n} c_k \varphi(p_k) = 0.$$

A clever choice for $p_1, \ldots, p_n$ will be done later.

For any $p \in \mathbb{R}^d$ and any rotation $R \in SO(d)$ we can write:

$$0 = \varphi(p)^T \sum_{k=1}^{n} c_k \varphi(p_k) = \sum_{k=1}^{n} c_k K(\|p - p_k\|) = \sum_{k=1}^{n} c_k K(\|Rp - Rp_k\|)$$

$$= \varphi(Rp)^T \sum_{k=1}^{n} c_k \varphi(Rp_k).$$

Since this is true for all $p' = Rp$ we can deduce that for all $p \in \mathbb{R}^d$ and all $R \in SO(d)$ we have:

$$\sum_{k=1}^{n} c_k k(p - Rp_k) = 0.$$

We denote by $\Lambda$ the finite measure on $\mathbb{R}^d$ given by Bochner's theorem applied on $k$.

Let us take a test function $\psi \in \mathcal{S}(\mathbb{R}^p)$ in the Schwartz space, we can write successively that for all rotation $R \in SO(d)$:

$$
\begin{aligned}
0 &= \int_{\mathbb{R}^d} \mathcal{F}\psi(p) \sum_{k=1}^{n} c_k k(p - Rp_k) dp \\
&= \int_{\mathbb{R}^d} \mathcal{F}\psi(p) \sum_{k=1}^{n} c_k \int_{\mathbb{R}^d} e^{i\omega \cdot (p - Rp_k)} d\Lambda(\omega) dp, \quad \text{(Bochner's theorem)} \\
&= \int_{\mathbb{R}^d} \left( \sum_{k=1}^{n} c_k e^{-i\omega \cdot (Rp_k)} \right) \int_{\mathbb{R}^d} \mathcal{F}\psi(p) e^{i\omega \cdot p} dp \, d\Lambda(\omega), \quad \text{(Fubini's theorem)} \\
&= (2\pi)^d \int_{\mathbb{R}^d} \psi(\omega) \sum_{k=1}^{n} c_k e^{-i\omega \cdot (Rp_k)} d\Lambda(\omega), \quad \text{(Fourier inversion)}
\end{aligned}
$$

As $K$ is not constant, we can find $\omega_0 \in \mathbb{R}^d \backslash \{0\}$ such that for all $\epsilon > 0$ small enough we have $\Lambda\big(B(\omega_0, \epsilon)\big) > 0$ (otherwise the finite positive measure $\Lambda$ would be concentrated on $0$ and $k$ would be constant).

Let $R \in SO(d)$, if we assume that $S := \sum_{k=1}^{n} c_k e^{-i\omega_0 \cdot (Rp_k)} \neq 0$ then we can find a small enough open ball $B(\omega_0, \epsilon)$ on which $Re(S))$ and $Im(S)$ have constant sign and such that: $|Re(S)| \geq c_1 > 0$ or $|Im(S)| \geq c_1 > 0$.

We choose $\psi$ such that $\psi \geq 0$, $\psi$ has compact support in $B(\omega_0, \epsilon)$ and $\psi \geq c_2 > 0$ on $B(\omega_0, \frac{\epsilon}{2})$. Then we obtain a contradiction by writing $0 \geq (2\pi)^d) c_1 c_2 \Lambda(B(\omega_0, \frac{\epsilon}{2}))$. (We separate real and imaginary parts).

This implies that:

$$\forall R \in \mathrm{SO}(d), \ \sum_{k=1}^{n} c_k e^{-i(R\omega_0) \cdot p_k} = 0, \tag{5}$$

Now we take a particular choice of $(p_i)$, let $p_k = (k, 0, \ldots, 0) \in \mathbb{R}^d$.

Up to rotations, we can assume without loss of generality that $\omega_0 = (w, 0, \ldots, 0)$ with $w \neq 0$. Moreover, we consider the particular case of rotations in the 2D plane generated by $(1, 0, \ldots, 0)$ and $(0, 1, 0, \ldots, 0)$.

Therefore, equation 5 implies that:

$$\forall \theta \in \mathbb{R}, \ \sum_{k=1}^{n} c_k \big( e^{-iw\cos(\theta)} \big)^k = 0,$$

So that the polynomial $\sum_k c_k z^k$ has an infinite number of roots. Thus $c_1 = \cdots = c_n = 0$.

### C.3 Random features embedding

In this section we give some details about the way we define random embeddings, which is very similar but slightly different than in [5].

If the kernel is properly scaled (i.e. $k(0) = 1$) then $\Lambda$ defines a probability measure. That's why we introduce a probability measure $\mathbb{Q}$ and write:

$$k(r) = k(0) \int_{\mathbb{R}^d} e^{i\omega \cdot r} d\mathbb{Q}(\omega) = k(0) \mathbb{E}_{\omega \sim \mathbb{Q}}[e^{i\omega \cdot r}].$$

Now, following the reasoning in [5] we consider:

$$\varphi(p)_i = \sqrt{2k(0)} \sin(\omega \cdot p + \frac{\pi}{4} + b)$$

With $\omega \sim \mathbb{Q}$ and $b$ a random variable with a symmetric law (note that $\mathbb{Q}$ is also symmetric). Then we have:

$$\mathbb{E}[\varphi(p)_i \varphi(p')_i] = 2k(0)\mathbb{E}\left[\left(\frac{e^{i\omega.p+\frac{\pi}{4}+b} - e^{-i\omega.p-\frac{\pi}{4}-b}}{2i}\right)\left(\frac{e^{i\omega.p'+\frac{\pi}{4}+b} - e^{-i\omega.p'-\frac{\pi}{4}-b}}{2i}\right)\right]$$

$$= -\frac{k(0)}{2}\left(e^{i\frac{\pi}{2}}\mathbb{E}[e^{i\omega.(p+p')+2b}] + e^{-i\frac{\pi}{2}}\mathbb{E}[e^{-i\omega.(p+p')-2b}]\right.$$

$$\left. - \mathbb{E}[e^{i\omega.(p-p')}] - \mathbb{E}[e^{-i\omega.(p-p')}]\right)$$

$$= k(0)\mathbb{E}[e^{i\omega.(p-p')}]$$

$$= k(p - p').$$

Therefore we reduce the variance by drawing i.i.d. samples $\omega_1, \dots, \omega_{n_0}$ and $b_1, \dots, b_{n_0}$ as described in section 3 and computing the mean $\frac{1}{n_0}\varphi(p)^T\varphi(p')$. By the strong law of large numbers we have the almost sure convergence:

$$\frac{1}{n_0}\varphi(p)^T\varphi(p') \xrightarrow[n_0 \to \infty]{} k(p - p'),$$

Now we can obtain Gaussian embedding by drawing the bias from $\delta_0$ and weights from $\mathcal{N}(0, \frac{1}{\ell^2}I_d)$. from the above formulas we immediately get:

$$k(p - p') = e^{-\frac{\|p-p'\|_2^2}{2\ell^2}}.$$

# D   Precise computations of the Neural Tangent Kernel

We now give more details about the computation of the limiting NTK and detail how we obtain the limiting kernels used in Figures 6 and 7 of the paper.

## D.1   Limiting NTK

For this purpose, following several authors ([3], [7], [4]), we need to introduce some gaussian processes and their associated kernels. For a symmetric positive kernel $\Sigma$ let us define:

$$\begin{cases} \mathcal{T}(\Sigma)(z, z') = \mathbb{E}_{(X,Y) \sim \mathcal{N}(0, \Sigma_{z,z'})}[\mu(X)\mu(Y)] \\ \dot{\mathcal{T}}(\Sigma)(z, z') = \mathbb{E}_{(X,Y) \sim \mathcal{N}(0, \Sigma_{z,z'})}[\dot{\mu}(X)\dot{\mu}(Y)] \end{cases} \quad \text{With}: \quad \Sigma_{z,z'} = \begin{pmatrix} \Sigma(z, z) & \Sigma(z, z') \\ \Sigma(z, z') & \Sigma(z', z') \end{pmatrix}.$$

Then we set $\Sigma^1(z, z') = \Theta^1_\infty(z, z') = \beta^2 + \frac{\alpha^2}{n_0}z^T z'$ and we define recursively:

$$\sigma^{l+1} = \beta^2 + \alpha^2 \mathcal{T}(\Sigma^l), \quad \dot{\Sigma}^{l+1} = \alpha^2 \dot{\mathcal{T}}(\Sigma^l), \quad \Theta^{l+1}_\infty = \dot{\Sigma}^{l+1}\Theta^l_\infty + \Sigma^{l+1}. \tag{6}$$

Using those formulas it is clear that the limiting NTK is invariant under rotation.

When neurons have constant variance, the following notion of dual activation function is often very useful:

**Definition D.1.** *Let $\mu : \mathbb{R} \longrightarrow \mathbb{R}$ be a function such that $\mathbb{E}_{X \sim \mathcal{N}(0,1)}[\mu(X)^2] < +\infty$, then its dual function $\hat{\mu} : [-1, 1] \longrightarrow \mathbb{R}$ is defined by:*

$$\hat{\mu}(\rho) = \mathbb{E}_{(X,Y) \sim \mathcal{N}(0, \Sigma_\rho)}[\mu(X)\mu(Y)], \quad \text{With}: \Sigma_\rho = \begin{pmatrix} 1 & \rho \\ \rho & 1 \end{pmatrix}.$$

We will use some properties of the dual function, which are described in [1].

## D.2 Another way of seeing Gaussian embedding

As explained above (Section 3.2.2), the Gaussian embedding can be seen as the first hidden layer of a neural network, with the first layer untrained. Thus it actually corresponds to $\Sigma^2$ with the above notations.

Let us consider the activation function $\mu : a \longmapsto \lambda \sin(\omega a + \frac{\pi}{4})$ and denote:

$$\forall x, y \in \mathbb{R}^{n_0}, \; \Sigma^1_{x,y} = \begin{pmatrix} \beta^2 + \dfrac{1-\beta^2}{n_0}\|x\|_2^2 & \beta^2 + \dfrac{1-\beta^2}{n_0}x^T y \\ \beta^2 + \dfrac{1-\beta^2}{n_0}x^T y & \beta^2 + \dfrac{1-\beta^2}{n_0}\|y\|_2^2 \end{pmatrix},$$

We are looking at:

$$\Sigma^2(x, y) = \beta^2 + (1-\beta^2)\mathbb{E}_{(X,Y)\sim\mathcal{N}(0,\Sigma^{(1)}_{x,y})}[\mu(X)\mu(Y)].$$

Let $(X, Y) \sim \mathcal{N}(0, \Sigma^1_{x,y})$, then $X - Y$ and $X + Y$ are normal random variables and $\mathbb{V}(X - Y) = \frac{1-\beta^2}{n_0}\|x - y\|_2^2$. Thus, using properties of characteristic functions we get:

$$
\begin{aligned}
\mathbb{E}[\mu(X)\mu(Y)] &= \lambda^2 \mathbb{E}\left[\left(\frac{e^{i\omega X+\frac{\pi}{4}} - e^{-i\omega X-\frac{\pi}{4}}}{2i}\right)\left(\frac{e^{i\omega Y+\frac{\pi}{4}} - e^{-i\omega Y-\frac{\pi}{4}}}{2i}\right)\right] \\
&= -\frac{\lambda^2}{4}\left(e^{i\frac{\pi}{2}}\mathbb{E}[e^{i\omega(X+Y)}] + e^{-i\frac{\pi}{2}}\mathbb{E}[e^{-i\omega(X+Y)}] - \mathbb{E}[e^{i\omega(X-Y)}] - \mathbb{E}[e^{-i\omega(X-Y)}]\right) \\
&= \frac{\lambda^2}{2}\mathbb{E}[e^{i\omega(X-Y)}] \\
&= \frac{\lambda^2}{2}\exp\left\{-\frac{1}{2}\omega^2\frac{1-\beta^2}{n_0}\|x - y\|_2^2\right\}.
\end{aligned}
$$

## D.3 Computation of the NTK used for Figure 7 in the paper

In this section we show how one can derived analytically the function $\Phi_\infty$ described in Section 4.3. This kind of computation can be used to derive numerically the filter radius $\hat{R}_{1/2}$ and tune the hyperparameters.

We use here a Gaussian embedding $\varphi$ of size $n_0$ with lenghtscale $\ell$ followed by one hidden linear layer (activation function $x \to \sqrt{2}\max(0, x)$) of size $n_1$ and the output layer $n_2 = 1$. We also take $\alpha^2 + \beta^2 = 1$ in those experiments, to ensure constant variance of the neurons.

By the strong law of large numbers we have for the limiting NTK of the first layer:

$$\Theta^1_\infty(\varphi(p), \varphi(p')) = \beta^2 + \frac{1-\beta^2}{n_0}\varphi(p)^T\varphi(p') \underset{n_0\to\infty}{\longrightarrow} \beta^2 + (1-\beta^2)e^{-\frac{\|p-p'\|_2^2}{2l^2}} =: G(\|p - p'\|).$$

For the second layer, we use the notion of dual function defined above. In the case of the standardized ReLu it is computed in [1]:

$$\hat{r}(\rho) = \rho - \frac{\rho\arccos(\rho) - \sqrt{1-\rho^2}}{\pi}, \quad \rho \in [-1, 1],$$

and:

$$\hat{\dot{r}}(\rho) = \dot{\hat{r}}(\rho) = 1 - \frac{\arccos(\rho)}{\pi}.$$

So that we can write, with $d = \|p - p'\|$:

$$\Phi_\infty(d) = \hat{r}(G(d)) + G(d)\dot{\hat{r}}(G(d)).$$

Therefore $\Phi_\infty$ only depends on $\ell$ and $\beta$. From this expression we can use standard Python libraries to approximate $\hat{R}_{1/2}$ for given values of the hyperparameters.

### D.4  Computation of the NTK used for Figure 6 in the paper

Now we derive an approximate of the quantity $\hat{R}_{1/2}$ used in Figure 6 of the paper. This is a little bit more difficult than with Gaussian embedding because the rotation invariance is now only an approximation, even in the infinite-width limit.

With Torus embedding, we have $n_0 = 4$. The embedding is followed by two hidden linear layers with standardised cosine activation function, and then the last linear layer. We used here $r = \sqrt{2}$ $\delta = \frac{\pi}{80}$ (which is the formula suggested in the paper with $n_x = n_y = 40$). As in the case of Gaussian embedding, we set $\alpha^2 = 1 - \beta^2$. This ensures that neurons have constant variance and allows easy analytical computations.

Thanks to the Torus embedding described above, we get for the first layer:

$$\Theta^1_\infty(\varphi(p), \varphi(p')) = \beta^2 + \frac{1 - \beta^2}{n_0}\varphi(p)^T\varphi(p')$$

$$= \beta^2 + \frac{1 - \beta^2}{2}\big(\cos(\delta(p_1 - p'_1)) + \cos(\delta(p_2 - p'_2))\big)$$

As rotation invariance is not analytically correct here, we look at the limiting NTK in the direction $p_1 = p_2$. which gives:

$$\Sigma^1(\varphi(p), \varphi(p')) = \Theta^1_\infty(\varphi(p), \varphi(p')) = \beta^2 + (1 - \beta^2)\cos(\delta r),$$

with $r = |p_1 - p'_1| = |p_2 - p'_2|$.

For the next layers, we use the dual function of the standardised cosine (see [1]) given by:

$$\hat{\mu}(\rho) = \frac{\cosh(\omega^2\rho)}{\cosh(\omega^2)},$$

and its derivative:

$$\dot{\hat{\mu}}(\rho) = \omega^2\frac{\sinh(\omega^2\rho)}{\cosh(\omega^2)},$$

Then the limiting NTK is simply given by the following formulas:

$$\Sigma^{l+1}(\varphi(p), \varphi(p')) = \beta^2 + (1 - \beta^2)\hat{\mu}(\Sigma^l(\varphi(p), \varphi(p'))),$$

$$\dot{\Sigma}^{l+1}(\varphi(p), \varphi(p')) = (1 - \beta^2)\dot{\hat{\mu}}(\dot{\Sigma}^l(\varphi(p), \varphi(p'))),$$

$$\Theta^{l+1}_\infty(\varphi(p), \varphi(p')) = \Sigma^{l+1}(\varphi(p), \varphi(p')) + \dot{\Sigma}^{l+1}(\varphi(p), \varphi(p'))\Theta^l_\infty(\varphi(p), \varphi(p')).$$

This way we construct a function $\Phi_\infty(r)$ with $r$ an approximation of the radius and we can use it to compute numerically an approximation of $\hat{R}_{1/2}$ as before.

## E  Square root of the NTK in the case of random embedding

We now prove that we can define a notion of a square root of the NTK. First we need a technical lemma:

**Lemma E.1.** *Let $\mu$ be a continuous function such that $\mathbb{E}_{X \sim \mathcal{N}(0,1)}[\mu(X)^2] = 1$, $C \in [0, 1]$ a constant and $f \geq 0$ a positive definite function (in the sense of definition C.1) such that $C + f(p) \leq 1$. Then the function*

$$F : p \longmapsto \hat{\mu}(C + f(p)) - \hat{\mu}(C),$$

*is positive definite, where $\hat{\mu}$ denotes the dual function of $\mu$ (see definition D.1).*

*Proof:*

Let us take $p_1, ..., p_m \in \mathbb{R}^d$ and $c_1, ..., c_m \in \mathbb{R}$. We introduce the Hermite expansion $\sum_k a_k h_k$ of $\mu$ and write its dual function as (see [1]):

$$\hat{\mu}(\rho) = \sum_{k=0}^{+\infty} a_k^2 \rho^k, \quad \rho \in [-1, 1],$$

Then by Bernoulli's formula:

$$\hat{\mu}(C + f(p_i - p_j)) - \hat{\mu}(C) = \sum_{k=1}^{+\infty} a_k^2 f(p_i - p_j) \sum_{s=0}^{k-1} C^{k-1-s}(C + f(p_i - p_j))^s.$$

Thus by polynomial combination with positive coefficients of positive semi-definite kernels:

$$\sum_{i,j=1}^{m} c_i c_j F(p_i - p_j) = \sum_{k=1}^{+\infty} \sum_{s=0}^{k-1} a_k^2 C^{k-1-s} \sum_{i,j=1}^{m} c_i c_j f(p_i - p_j)(C + f(p_i - p_j))^s \geq 0,$$

Which achieves the proof.

Let us recall the statement that we want to prove:

**Proposition E.1** (Proposition 3.4 in the paper). *Let $\varphi$ be an embedding as described in section 3.2.2 of the paper, for a positive radial kernel $k \in L^1(\mathbb{R}^d)$ with $k(0) = 1$. Then there is a filter function $g : \mathbb{R} \to \mathbb{R}$ and a constant $C$ such that for all $p, p'$:*

$$\lim_{n_0 \to \infty} \Theta_\infty(\varphi(p), \varphi(p')) = C + (g \star g)(p - p'), \tag{7}$$

*where $\Theta_\infty$ is the limiting NTK of a network with Lipschitz, non constant, and standardized activation function $\mu$.*

Before writing the proof, let us make some remarks on the assumptions of this proposition and their immediate implications:

- We recall that the fact that $\mu$ is "standardised" means here: $\mathbb{E}_{X \sim \mathcal{N}(0,1)}[\mu(X)^2] = 1$.

- As mentioned before (Section 2.3 of the paper) we assume for simplicity that $\alpha^2 = 1 - \beta^2$ to ensure constant variance of the neurons (we consider $\beta \in [0, 1)$).

- We denote by $A$ the Lipschitz constant of $\mu$. By Rademacher theorem, we know that $\mu$ is almost everywhere differentiable and $\|\dot{\mu}\|_\infty \leq A$. The fact that $\mu$ is not constant ensures that $\hat{\mu}$ is (strictly) increasing on $[0, 1)$.

- Moreover, the Lipschitz assumption also implies that $|\hat{\mu}(1)| \leq A^2 < +\infty$ and therefore $\hat{\mu}$ is continuous on $[-1, 1]$ by Abel's theorem on entire series.

- The procedure to approximate the kernel $k$ in Section 3.2.2 of the paper assumes that $k$ is continuous (to be able to apply Bochner's theorem). It is therefore also the case in this proof.

*Proof of the proposition:*

**Step** 1: We want to show by recursion that for all $l \geq 1$ there exists some constant $C_l \in [0, 1)$ such that for all $p, p' \in \mathbb{R}^d$ we have in probability:

$$\Sigma^l(\varphi(p), \varphi(p')) \xrightarrow[n_0 \to \infty]{} C_l + f_l(p - p'), \tag{8}$$

With $f_l$ a radial positive definite function such that $f_l \geq 0$ and $f_l \in L^1(\mathbb{R}^d)$.

For $l = 1$, we know that this is true by the law of large numbers:

$$\Sigma^1(\varphi(p), \varphi(p')) = \Theta_\infty^1(\varphi(p), \varphi(p')) = \beta^2 + \frac{1 - \beta^2}{n_0} \varphi(p)^T \varphi(p')$$

$$\xrightarrow[n_0 \to \infty]{} \beta^2 + (1 - \beta^2)k(p - p'), \tag{9}$$

We just set $f_1 = (1 - \beta^2)k$. Now we assume $l \geq 2$:

We have by our normalisation assumptions $\Sigma^l(\varphi(p), \varphi(p)) = C_l + f_l(0) = 1$. Using the continuity of $\hat{\mu}$ (see [1] for the properties of $\hat{\mu}$), we have:

$$\Sigma^{l+1}(\varphi(p), \varphi(p')) = \beta^2 + (1 - \beta^2)\hat{\mu}(\Sigma^l(\varphi(p), \varphi(p')))$$

$$\xrightarrow[n_0 \to \infty]{} \beta^2 + (1 - \beta^2)\hat{\mu}(C_l + f_l(p - p')). \tag{10}$$

Using properties of the dual function given in [1], we know that $\hat{\mu}$ is positive, increasing and convex in $[0, 1]$. Moreover as $f_l$ is radial positive definite we have $f_l \leq f_l(0) = 1 - C_l$. Then by convexity:

$$\hat{\mu}(C_l + f_l(p - p')) = \hat{\mu}\left(\frac{f_l(p - p')}{1 - C_l} + \left(1 - \frac{f_l(p - p')}{1 - C_l}\right)C_l\right)$$
$$\leq \frac{f_l(p - p')}{1 - C_l}\hat{\mu}(1) + \left(1 - \frac{f_l(p - p')}{1 - C_l}\right)\hat{\mu}(C_l).$$

Using that $\hat{\mu}$ is increasing:

$$|\hat{\mu}(C_l + f_l(p - p')) - \hat{\mu}(C_l)| \leq \frac{\hat{\mu}(1) - \hat{\mu}(C_l)}{1 - C_l}f_l(p - p'),$$

So that we can rewrite equation 10 in the following form:

$$\Sigma^{l+1}(\varphi(p), \varphi(p')) \xrightarrow[n_0 \to \infty]{} \beta^2 + (1 - \beta^2)\hat{\mu}(C_l) + f_{l+1}(p - p'),$$

With $f_{l+1}(p - p') = (1 - \beta^2)(\hat{\mu}(C_l + f_l(p - p')) - \hat{\mu}(C_l))$ and $C_{l+1} = \beta^2 + (1 - \beta^2)\hat{\mu}(C_l)$.

The previous inequality, lemma E.1 and the fact that $\hat{\mu}$ is increasing in $[0, 1)$ ensure the properties of $f_{l+1}$ and $C_{l+1}$.

**Step 2:** As $\hat{\mu}$ is also positive, continuous, increasing and convex in $[0, 1]$, we can obtain a convergence in probability similar to equation 8 but for $\dot{\Sigma}^l$:

$$\dot{\Sigma}^l(\varphi(p), \varphi(p')) \xrightarrow[n_0 \to \infty]{} B_l + h_l(p - p'),$$

With $B_l \geq 0$, and $h_l$ a positive definite function such that $h_l \in L^1(\mathbb{R}^d)$ and $h_l \geq 0$.

Now we want to show by recursion that for a fixed $l$:

$$\forall p, p' \in \mathbb{R}^d, \ \Theta^l_\infty(\varphi(p), \varphi(p')) \xrightarrow[n_0 \to \infty]{} C_{\mu,\beta,l} + \theta_l(p - p'). \tag{11}$$

With $\theta_l$ a positive definite function such that $\theta_l \in L^1(\mathbb{R}^d)$ and $C_{\mu,\beta,l} \geq 0$. Again we know that this is true for $l = 1$ by equation 9.

We have:

$$\Theta^{l+1}_\infty(\varphi(p), \varphi(p')) \xrightarrow[n_0 \to \infty]{} (C_{\mu,\beta,l} + \theta_l(p - p'))\dot{\Sigma}^{(l+1)}(p, p') + C_{l+1} + f_{l+1}(p - p').$$

So that we can set:

$$\theta_{l+1}(\varphi(p), \varphi(p')) = C_{\mu,\beta,l}h_{l+1}(p - p') + \theta_l(p - p')\dot{\Sigma}^{l+1}(\varphi(p), \varphi(p')) + f_{l+1}(p - p'),$$

and:

$$C_{\beta,\mu,l+1} = C_{l+1} + C_{\beta,\mu,l}B_l.$$

Using that $|\theta_l(p - p')\dot{\Sigma}^{l+1}(\varphi(p), \varphi(p'))| \leq A^2|\theta_l(p - p')|$ and all the previous results, the recursion works automatically and we have equation 11 for all $l \geq 2$.

Moreover $(p, p') \longmapsto \theta_l(p - p')\dot{\Sigma}^{1+l}(p, p')$ is positive semi-definite as a product of two positive semi-definite kernels. By sum we deduce that $\theta_{l+1}$ is positive semi-definite and by recursion we have the result for all $\theta_l$.

**Step 3:** Now, using integrability of $\theta_l$, we know that its Fourier transform defines a function $q \in L^\infty(\mathbb{R}^d)$.

From dominated convergence theorem we deduce that $q$ is continuous.

Therefore in the sense of distributions, the Fourier transform of $\theta_L$ is given by a finite positive measure (Bochner's theorem) and also by $q \in L^\infty(\mathbb{R}^d)$. We deduce that $q$ is the density of this finite positive measure (the Radon-Nikodym derivative with respect to the Lebesgue measure).

From those arguments we get $q \geq 0$ and $q \in L^1(\mathbb{R}^d)$. We then have the Fourier inversion formula for $\theta_L$:

$$\theta_L(p - p') = \frac{1}{(2\pi)^d} \int_{\mathbb{R}^d} q(\omega) e^{i\omega.(p-p')} d\omega, \quad \text{with: } q \geq 0$$

Hence it makes sense to define:

$$g = \mathcal{F}^{-1}(\sqrt{q}),$$

In the sense of the Fourier transform of a $L^2$ function. Then the convolution theorem ensures:

$$\theta_L = g \star g.$$

**Remark:** Here we used lemma E.1 and the dual activation function to show that both $f_l$ and $\theta_l$ are positive definite. If we only show that $\theta_l \in L^1(\mathbb{R}^d)$ it is still possible to show the same properties of the function $q$ by using positive definiteness of $C + \theta_L$ and take the Fourier transform in the sense of distributions, which leads to $(2\pi)^d C \delta_0 + q = (2\pi)^d M$ with $M$ a finite positive measure. Then arguments based on test functions and the continuity of $q$ give the result. The advantage of lemma E.1 is that it is a bit more general.

# F   Additional experimental results

## F.1   Plots of the Neural Tangent Kernel

Here are some additional experimental results regarding the comparison between the theoretical (limiting) NTK $\tilde{\Theta}_\infty$ and the empirical NTK $\tilde{\Theta}_{\theta(t)}$. Here again the "lines" of the Gram matrices are reshaped as images.

Figure 1 represents the comparison between the limiting NTK and the emprirical NTK with a Gaussian embedding. We can observe that the infinite-width limit seems to be well-respected.

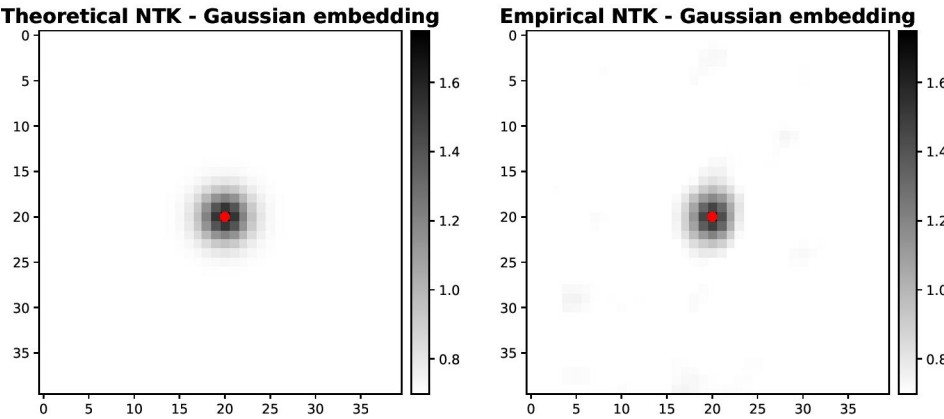

Figure 1: Comparison between one line of the Gram matrix of the empirical NTK $\tilde{\Theta}_{\theta(t)}$ and and of the corresponding limiting NTK $\tilde{\Theta}_\infty$. Here we use a Gaussian embedding as described in the paper

Figure 2 shows the evolution of the NTK during the optimisation process. While the NTK begins to change at the end of training (it is due to the alignment of descent directions, because of the sigmoid we use to control the volume, pre-densities $(x_i)_{1 \leq i \leq N}$ tend to infinity) the NTK stays close to $\Theta_\infty$ during the part of training where the final shape is created. This justifies even more that it is pertinent to study the effect of the NTK on the final geometry.

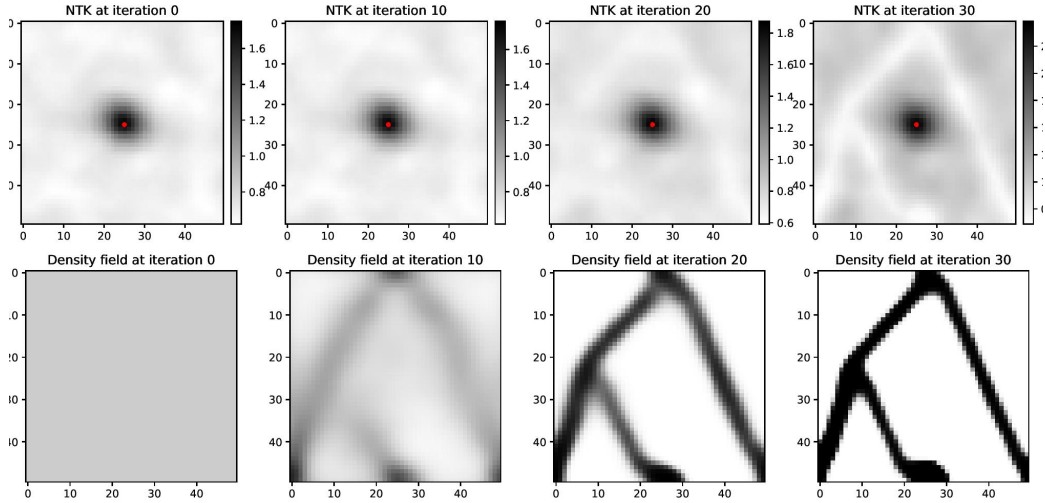

Figure 2: Evolution of the NTK of a network with a Gaussian embedding with hyperparameters as described in Section 4.1. We can see a relative stability of the NTK