# OpenReview forum: "DNN-based Topology Optimisation:  Spatial Invariance and Neural Tangent Kernel"
_NeurIPS.cc/2021/Conference — NeurIPS 2021 Poster_

### Official Review · Reviewer_r379 · 2021-07-05

**Rating:** 6
**Confidence:** 4

**Summary:**

The paper introduces a deep fully-connected neural network (FCNN) approach to topology optimization, in which a neural network learns to predict density as a function of spatial location, with gradient feedback based on Solid Isotropic Material Penalization (SIMP). The authors draw a connection between the NTK Gram matrix of this FCNN and the spatial low-pass filtering usually done in non-neural topology optimization, and propose two input embeddings that induce shift-invariance in the NTK. Instead of tuning the filtering radius of an explicit low-pass filter, with the proposed FCNN approach the effective filtering can be tuned by adjusting parameters of the embedding and the network architecture to vary the width of the NTK and control its spectrum.


**Limitations And Societal Impact:**

The paper would benefit from a discussion of limitations.

**Main Review:**

Overall, the paper is clear and interesting. The authors demonstrate that their proposed embeddings induce shift-invariance and the ability to tune the bandwidth of the resulting NTK. These adjustments to the embedding (and in some cases, network architecture) translate to interpretable changes in the resulting optimized shapes, and a marked improvement over an FCNN without an input embedding. However, I have some questions/concerns, especially regarding prior work and limitations, that make me question how novel and how useful the proposed approach really is.

Prior work:

The discussion of prior work is limited to the introduction, and focuses almost exclusively on the problem of topology optimization. There has been a wealth of recent progress on different problems e.g. in graphics, in which the goal is again to model a structure, using a FCNN to map spatial coordinates to density (and sometimes other properties). These works also make use of an input embedding (e.g. https://arxiv.org/abs/2006.10739) and/or a periodic activation function (e.g. https://arxiv.org/abs/2006.09661), quite similar to the proposed Gaussian random Fourier feature embedding and torus embedding with cosine activations, respectively. Although to my knowledge the application of this technique to the problem of topology optimization is novel, the embeddings themselves are less so.

Additionally, improvement is only shown with respect to a FCNN with no embedding; no other prior techniques are compared against. Is there a benefit to using the proposed approach instead of e.g. the classical filtering method?

Limitations/other questions:

Although the experiments are compelling, they are limited to two-dimensional problems. Is this standard in topology optimization? An experiment in three dimensions would be even more convincing.

What are the strengths and weaknesses of each of the two proposed embedding methods? Why introduce both unless there are some situations where either is preferable to the other?

In section 1.1 (first bullet point), doesn’t the equivalence to the NTK also require an assumption of gradient flow (infinitesimal learning rate), in addition to the assumption of infinite width?

If the torus embedding uses a periodic activation function, how can you separate the effects of using the embedding from the effects of the activation function?


Minor/low-level comments/questions:

SIMP is mentioned in the abstract, without being defined until the introduction.

In a few places (equations 2, 5, and 6 in particular), a formula is introduced with no explanation or derivation. Although derivations are available in the supplement, it would be helpful to have a bit more intuition/explanation provided in the main text.

Line 72: An equality condition on the sum of the densities is described as a volume constraint. Would it be more accurate to call this a mass constraint?

Line 83: The distinction between X and \bar X is not clear.

Line 87-88: This might be more clearly worded by reversing the order, i.e. D_X is PSD with all eigenvalues less than ½, and its nullspace is… (especially since the word “kernel” is a bit overloaded).

Line 91: Having a scalar output is a feature of the FCNNs used in this paper, but not a general property of all FCNNs (as is implied).

Figure 1: The figure and caption use X and Y, but the surrounding text includes superscripts NN. Are these the same quantities?

Line 138-139: Do the z’s need to be normalized for this to hold?

Line 149: Where does the 4-norm in the error term come from?

Last line on page 6: Why the need to add pi/4? Isn’t this subsumed by b_i?

Equation 9: What does star denote?

Typos:

Line 63-64: comma splice

Line 77: “it can used” -> “it can be used”

Figure 4 caption: Left and right are reversed

Line 195: “lenghtscale” -> “length scale”

Line 238: “hyperparameters network” -> “hyperparameters of the network”

Line 243: “illustrate” -> “illustrates”

Line 255, 257: bad figure reference (should be figures 8, 9)

**Time Spent Reviewing:**

3

---

> ### Author Response · Authors · 2021-08-04
> **Author's Answer**
>
> Prior Work:
> Thanks for the two references, they are indeed closely related. We will discuss their relation to the embeddings we propose. Since the publication we came aware of some other papers proposing similar techniques but none in the context of topology optimization.
>
> Comparison to prior techniques:
> The goal of this paper was principally to give a theoretical description of topology optimization with DNNs in the NTK regime, and the resulting insights on the role of the hyper-parameters of the network and less to obtain the "best" algorithm for topology optimization. Note also that the evaluation of topology optimization is in part subjective: the goal is not only to reduce the compliance but also to obtain a shape without checkerboard patterns for example.
>
> 3d topology optimization:
> Our theoretical analysis does not depend on the input dimension, we therefore do not expect any big difference in behavior for the 3d case.
>
> Choice of Embedding:
> The torus embedding is simpler and requires only a small embedding dimension, at the cost of getting only an approximate rotation invariance. In settings where the rotation invariance is not crucial, it may be best to use the torus embedding. On the other hand, the Gaussian embedding has the advantage of making it easy to control the filter size.
>
> Gradient Flow:
> While our analysis is done for gradient flow, there exists a number of papers showing the existence of the NTK limit also for gradient descent (for example https://arxiv.org/abs/1902.06720v1). There is no reason to expect the topology optimization setting to be any different.
>
> Torus Embedding / Sinus activation:
> When used in combination, the role of the torus embedding is to ensure translation/rotation invariance and the role of the sinus activation is to reduce the filter width. As you point out these two roles are not completely orthogonal as the the scale of the grid in the embedding can also be used to change the filter width, however without the embedding the sinus activation is not able to guarantee the rotation/translation invariance.
>
> Thanks for pointing out to minor problems/typos, we will correct them.

---

### Official Review · Reviewer_cu3P · 2021-07-16

**Rating:** 6
**Confidence:** 2

**Summary:**

The paper studies topology optimization with neural networks. In particular, they use Deep neural networks to generate the density at each point used by Solid Isotropic Material Penalisation (SIMP) methods. The deep neural network has the input as the coordinates and the output is density (the amount of material) at that coordinate.



**Limitations And Societal Impact:**

Yes

**Main Review:**



1) In equation 1, how is it guarantees that there is a fix bias terms $\bar{b}$ for all the elements in the grid that would satisfy the conservation of $V_{0}$ for any arbitrary $x_{i}$'s and $V_{0}$? Is it the reasoning for that is line 13 of your supplement? I think it would be good to refer in the main text that you answer to that question in the supplement. Also don't you need a condition that $V_{0} \leq N$? In the proof you used $\bar{b}$ as both the unique bias and also the graph function that is the consequence of Implicit Theorem, it is better to different this two notions.


2) Proposition 2.1 proof: I don't see why $F(\lambda) = 0$ only happens in open intervals $(a_{j},a_{i})$? Furthermore, how does this prove that the eigenvalues are between $[0,0.5]$?


3) In the experimental setup how did you choose the parameters for your neural network and the embedding? is there any sort of cross-validation involved?


4) Could've you not fix that translational invariancy via using equivariant networks, such networks exist, "Scaling-Translation-Equivariant Networks with Decomposed Convolutional Filters", it would've been nice to see the result in comparison with NTK?

**Time Spent Reviewing:**

6

---

> ### Author Response · Authors · 2021-08-04
> **Author's Answer**
>
> Thanks for your review and remarks.
> 1. We will improve the discussion of the bias term in the main. You are right that we need to assume $0\leq V_0 \leq N$ for this to make sense, we will mention it. We will also disambiguate the notation of the bias.
>
> 2. We will add more details in the Appendix regarding your questions.
>
> 3. We did not use cross-validation, since the goal of this paper is to describe the effect of the hyper-parameters of the network and less to obtain the best possible results. Note also that the objective in topology optimization is in part subjective, the goal is not only to reduce the compliance but also to obtain a shape without checkerboard patterns for example.
>
> 4. Thanks for your interesting link. One could indeed use such an approach if the density field was generated as in DCGANs, creating the whole image in one pass, in our case we generate the image as a function taking 2d coordinates as inputs and outputing a scalar, the problem of translation and rotation invariance in these two settings is very different. A similar analysis (for example using the NTK) could be done for this one-pass generation setting, but it is outside the scope of this paper.

---

> > ### Comment · Reviewer_cu3P · 2021-09-01
> > **Provide Details to question number 2**
> >
> >
> > I've not convinced by proposition 2.1 proof and your response was not enough to assure me of the validity of the proof.

---

> > > ### Author Response · Authors · 2021-09-02
> > > **Details to question number 2**
> > >
> > > In the proof we emphasize the fact that $a_i = \dot{\sigma}(x_i + \bar{b}(X))$, where $\sigma$ is the sigmoid function.
> > >
> > > Furthermore we changed the end of the proof to this to make it clearer:
> > >
> > > Eigenvalues: We already know that $0$ is an eigenvalue with multiplicity $1$. So let $u \neq 0$ in $\mathbb{R}^N$ and $\lambda > 0$ such that: $D_X u = \lambda u$. Then we easily show:
> > >     $$
> > >     \forall i \in1,\dots,N ,~~ \frac{a_i - \lambda}{a_i}u_i = \frac{1}{\vert \dot{S} \vert_1} \sum_{j=1}^N a_j u_j =: \langle u \rangle_a.
> > >     $$
> > >     If $\langle u \rangle_a = 0$, then necessarily $\lambda \in \{ a_1,...,a_N \}$
> > >     \\
> > >     If $\langle u \rangle_a \neq 0$, then we can assume (by normalising $u$) that $\langle u \rangle_a = 1$ and we have $u_i = \frac{a_i}{a_i - \lambda}$. Then we can replace $u_i = \frac{a_i}{a_i - \lambda}$ in the equation $\langle u \rangle_a = 1$:
> > >     $$
> > >      \sum_{j=1}^N a_j  =  \sum_{j=1}^N \frac{a_j^2}{a_j - \lambda}, \quad \text{which by substraction leads to} \quad F(\lambda) := \sum_{j=1}^N  \frac{a_j}{a_j - \lambda} = 0,
> > >     $$
> > >     By studying the function $F$, we see that $\forall \lambda > \max_i(a_i),~ F(\lambda) < 0$. Therefore an eigenvalue always satisfies the inequality:
> > >     $$
> > >     \lambda \leq \max \{ a_1,...,a_N\} \leq \Vert \dot{\sigma} \Vert_{\infty} = \frac{1}{4},
> > >     $$
> > >     The last inequality coming from $a_i = \dot{\sigma} (x_i + \bar{b}(X))$.
> > >
> > > We hope that it is clearer for you. Note that the bound on the eigenvalues is not required for the other results in the paper (and the bound is actually better than $\frac{1}{2}$). We present it merely to give intuition about the form of this matrix.

---

### Official Review · Reviewer_XBF1 · 2021-07-19

**Rating:** 6
**Confidence:** 3

**Summary:**

Built upon the NTK, this paper provides a theoretical understanding of using deep neural networks in the SIMP method for topology optimization in the infinite-width regime. The authors show that: 1) Topology optimization with a DNN is equivalent to topology optimization with the square root of NTK as the density filter. 2) They also propose two embeddings to ensure the spatial invariance of the NTK. 3) They study how the choice of embedding, activation function, depth, etc., affects the radius of the NTK.


**Limitations And Societal Impact:**

The limitation was discussed in the paper. The authors did not discuss the negative social impact considering that their work is theoretical.


**Main Review:**


This paper provides a theoretical understanding of using DNNs in the SIMP method for topology optimization and offers practical guidance on choosing the activation function, the choice of embedding, the depth of the neural networks for adjusting the radius of the NTK. The results in the paper can be good references for practitioners working in this field. However, the theoretical analysis primarily builds on the NTK theories developed in [11], limiting the theoretical novelty. Conversely, the proposed two embeddings for inducing spatial invariance are novel, to my knowledge. These two embeddings can also be useful for other areas where spatial invariance is required.  In the meantime, I have the following questions:


1) Since the approximation error for the torus embedding depends on the grid size, I wonder how to choose the grid size in practice? Also, how does the approximation error propagate? Will the later steps (e.g., the pointwise FCNN) amplify it?


2) For the two embeddings, the torus embedding, and the Gaussian embedding, what's the advantage of each other?  It would be great to discuss this and provide suggestions on the choice of embedding for some specific scenarios. By doing so, it can better relate the theories to practical problems in topology optimization.


I am not familiar with the research in topology optimization, so my assessment is based on the theoretical analysis in the paper. Overall, I am neutral to this paper but more on acceptance due to the above reasons.

====Post rebuttal=====

Thanks for the response! I would suggest authors include these discussions in the appendix. My score remains unchanged.

**Time Spent Reviewing:**

6

---

> ### Author Response · Authors · 2021-08-04
> **Author's answer**
>
> Thanks for your review. Regarding your questions:
> 1. The choice of the grid size leads to some kind of trade-off: for large grid sizes we get a kernel which is "less" rotation invariant with a small filter width, and for small grid sizes we get more rotation invariance but a wider filter. In our experiments, we obtained best results with small filter widths and observed that the translation invariance has a stronger effect on the resulting shape than rotation invariance (note that in traditional topology optimization, square filters are sometimes used, which are translation but not rotation invariant). For these reasons we chose the grid size to cover roughly half of the torus (note that taking the full torus leads to cyclic boundary conditions which is in general not needed). We can discuss this trade-off in the final version of the main.
>
> The effect of depth is more complex, and we have no simple answer regarding its effect.
>
> 2. The torus embedding is simpler and requires only a small embedding dimension, at the cost of getting only an approximate rotation invariance. In settings where the rotation invariance is not crucial, it may be best to use the torus embedding. On the other hand, the Gaussian embedding has the advantage of making it easy to control the filter size.

---

### Decision · Program_Chairs · 2021-09-27

**Decision:**

Accept (Poster)

**Comment:**

This paper considers the application of deep neural networks to the "SIMP" method for topology optimization. It proposes a novel approach for doing this and analyzes it using NTK theory.

This application is well outside the field of expertise of the reviewers, or indeed probably everyone associated with NeurIPS, including myself. That being said, the reviewers seem to think this is a good well-written contribution, with theoretical insights that might generalize beyond this particular application.